# Stressor-Induced Temporal Cortisol Deficiency as a Primary Trigger for Adaptation to Stress

**DOI:** 10.3390/ijerph19095633

**Published:** 2022-05-05

**Authors:** Ewa Latour, Jarosław Arlet, Emilia Latour, Marianna Latour, Piotr Basta, Anna Skarpańska-Stejnborn

**Affiliations:** Faculty of Physical Culture in Gorzów Wlkp., Poznan University of Physical Education, Ul. Estkowskiego 13, 66-400 Gorzów Wielkopolski, Poland; jarlet@int.pl (J.A.); latouremilia@gmail.com (E.L.); marianna.latour@o2.pl (M.L.); maly197@interia.pl (P.B.); ankass@poczta.onet.pl (A.S.-S.)

**Keywords:** stress, cortisol, fat percentage, adaptation, hypothalamic–pituitary–adrenal (HPA) axis

## Abstract

Background: Inconsistencies in measurements of cortisol response to stress have caused disagreements in the direction of the change in cortisol concentrations immediately after the onset of stress. Researchers typically observe increased cortisol levels in response to a stressor, perceiving occasional decreases as a sign of possible disorders. Reports indicate the relative ease of standardizing a physical stressor compared with a mental stressor, and cross-stressor adaptation is observable only in elite athletes. Methods: We investigated the cortisol response to top-intensity physical exertion by analyzing the course of the cortisol response, the changes in this response resulting from adaptation to intense exercise, and the possible convergence between the cortisol changes and body fat content. We examined 16 male athletes, members of the Polish National Rowing Team, competing in the World Rowing Championships, in top form, of an average training experience of seven years. The measurements were performed before and after the training camp preparatory to the Championships. We performed the measurements before and after the training camp preparatory to the Championships. Results: Before the camp, the athletes consistently reacted to the exertion test with a decrease in cortisol concentration and elevated cortisol levels after rest compared with baseline. After the camp, the post-exertion cortisol decrease as well as the post-rest cortisol elevation was much smaller and less consistent. Conclusions: The transient decrease in cortisol concentration at the onset of stress thus represents a physiological reaction, and the stress response counteracts the resulting cortisol deficiency to support cortisol availability during stress. Adaptation to stress enhanced this counteracting effect by (1) increasing the baseline cortisol concentration and (2) speeding up the response to its decline. This enhanced effect was boosted by adipose tissue.

## 1. Introduction

Stress accompanying adaptation to environmental demands triggers corticosteroid secretion by the adrenal cortex. This secretion takes place under the control of the hypothalamic–pituitary–adrenal (HPA) axis regulatory system [1,2], which ensures that corticosteroid secretion levels are maintained within safe limits. When conducting studies on stress-related issues, researchers have primarily observed increased cortisol levels in response to a stressor; thus, they tend to view other response patterns as causes or symptoms of somatic or psychological dysfunctions [3]. By contrast, other studies have reported decreased cortisol levels in response to both physical [4] and psychological stressors in healthy individuals without signs of impaired stress tolerance, although a decline in attention was observed in one case [5]; however, an improvement in cognitive functioning (behavioral and electrophysiological) was observed in another [6]. The authors of further studies attributed the decrease in cortisol levels to the failure to meet the measurement conditions of the experiment conducted [7,8]. Although ongoing research on adaptation has analyzed changes in reactive increases in cortisol levels, the declines of its concentration in the immediate response to a stressor have not yet become the focus of separate research, perhaps because of the small number of such cases reported. Thus, the dilemma of whether the decline in cortisol level indicates an inadequate or abnormal response to prolonged or excessive stress exposure or a normal, physiological reaction remains unclear.

Difficulties in understanding how cortisol concentration changes in response to stress arise because this response depends on both the type and level of the stressor [9]. Efforts to address these difficulties stem from the prevailing view of the co-occurrence of deteriorated sensitivity of the HPA axis involved in the management of cortisol stores and various functional impairments [10]. A weakened cortisol response to stress occurs both in disorders of psychological origin, such as post-traumatic stress disorder (PTSD), depression [11], and alcohol dependence [12], as well as in somatic or physical disorders, such as overtraining syndrome (OTS) in athletes. Acute cortisol support at the onset of stress results in reducing fatigue and improving vigor [13]. Hence, researchers who treat various illnesses are interested in shaping the cortisol regulatory system, specifically as it relates to a concept based on the not yet fully understood phenomenon of “cross-stressor adaptation”, which relies on gaining mental resilience through adaptation to a physical stressor [14,15]. However, studies on stress-induced cortisol reactions to stressors in these disorders have reported conflicting results [16]. Thus, identifying the hitherto unknown factors controlling the HPA axis could greatly improve the quality of assessment of its functioning regarding resilience and coping with stress [17]. However, the preservation of cortisol availability even after experimental destruction of the hypothalamus in rodents [18] shows greater complexity of the cortisol regulatory system: the HPA axis does not appear to be the sole controlling structure. This complexity of the cortisol regulatory system can be an important reason for the inconsistent and sometimes contradictory results of studies analyzing cortisol changes to understand adaptive disorders. Research demonstrating various reactive changes in cortisol concentrations accompanied by a comprehensive description of their determinants thus holds great value to researchers developing mathematical models to explain this complexity [19,20].

The reliability of the experimental assessment of the cortisol response to a stressor would benefit from the use of a physical exertion stimulus, because this would allow easier standardization compared with a pure psychological stimulus. It would also facilitate the selection of test participants with homogenous physical efficiency characteristics in terms of response to such a stressor. For example, one could select individuals uniquely adapted to cope with high levels of physical exertion. Given their habituation to such stressors and the relatively low probability of pathological responses, elite athletes are the most suited to assess adaptive changes in cortisol concentrations under stress. Additionally, the effectiveness of cross-stressor adaptation is observable only in elite athletes [21], supporting the rationale of selecting a study population of such athletes to test exertional stressor-stimulated responses. Regardless, no consensus exists on the course of response to an exertion stressor. According to a common assumption, from the onset of a high-intensity stress, the cortisol concentration in the blood begins to rise sharply [22], although many studies [4,6,23,24] show the opposite. The cortisol response to physical exertion undoubtedly undergoes influences from many factors related to both the functional and structural properties of the individual, such as age [25], physical capacity [26], and body build and composition (particularly differences in body fat content, which reportedly determine, to some extent, cortisol levels and their changes under stress [6]). However, none of these factors alone determines the post-exertion cortisol concentration; hence, even studies with similar participant groups and measurement protocols find divergent (both increasing and decreasing) trends in early response to exertion [16]. Thus, it seems that solely systematizing experiments regarding the determinants of the stressor response does not suffice to avoid inconsistencies in research results.

Discrepancies in study results may also stem from differences in experimental designs and measurement protocols used. These may involve, for example, assuming too long a time between the start of exposition to stress and the first measurement (typically around 30–60 min) compared with the time of the HPA axis response activation. These may also result from low-strength stressors that do not exceed the level at which the body can sustain cortisol levels from its current stores, and thus, small changes may go unnoticed because of the lag in the start of recording the signal in an adopted measurement method. Moreover, in studies exploring building resilience to stress, discrepancies in the assessment of changes in the cortisol concentration course in response to a stressor may result from differences in the properties of the training applied, such as its intensity, total duration, and type [27]. The above issues inspired us to investigate the cortisol response to a severe stressor of a quickly exhaustive, top-intensity physical exertion, specifically regarding three aspects: (1) the course of the cortisol response; (2) the changes in this response resulting from adaptation to intensive exercises repeated over a long period; and (3) possible convergence between the cortisol level or its changes and body fat content.

Cortisol deficiency at the onset of stress may result from a temporary inability to maintain a current cortisol level due to both an insufficient rate of cortisol production or release from existing stores and a lag time in the activation of its release. The feedback loop probably reacts to this deficiency also with a subsequent excessive increase of the cortisol level above baseline. Because a cortisol deficiency seems to be one of the earliest manifestations of the response to a stressor, we assume that it may be an essential component of the error signal for the feedback loop. If so, minimizing the delay in the onset of the release of cortisol stores or increasing the release rate to counteract this deficiency should thus underlie the adaptation to stress. Hence, we hypothesized that the adaptation results in a reduction in both the deficiency of cortisol concentration just after the onset of stress and its subsequent compensatory excess.

To verify the hypothesis, we analyzed how adaptation to a change in the organization of training routines modified the course of cortisol concentration in response to the strenuous maximal intensity exertion, which led to severe performance exhaustion in a time short enough to observe a possible early decrease in the cortisol level in immediate response to this stimulus. We examined a group of elite endurance athletes, the only group able to perform such an extreme effort in terms of intensity and exhaustion and, thus, most suitable for observing the early changes resulting from severe stress. Our study group of athletes was habituated to this particular exertion stress test and the accompanying sequence of measurements because they undergo them regularly as a standard for performance control, typically at the beginning and at the end of training cycles. Athletes develop an adaptation to exercise stress in the course of cyclic training camps, during which they train under a strict regime of systematic repetition of fitness exercises with shorter rest periods than they are accustomed to. Although the shortened rest periods could expose athletes to overtraining, most of them complete the training camp and compete in subsequent competitions in top form. This ensures that any changes observed in their functioning during the training camp are an outcome of the adaptation rather than overtraining.

## 2. Materials and Methods

### 2.1. Study Group

The participants were members of the Polish Rowing National Team. They were selected for the Team based on their performance in national rowing competitions and the results of performance tests corresponding to the rowing competitions. After a preparatory training camp, the national team members competed at the World Rowing Championships in peak form. The study group comprised the members who agreed to take measurements related to the research. Their mean training experience was 7 (SD = 1.71) years.

### 2.2. Experiment Conditions

The study was conducted between March and May 2018 during a long-term training camp scheduled between the preparatory and competitive phases of the yearly training cycle. This camp was characterized by workouts of high regularity and high-intensity exercises. More detailed characteristics of the training are presented in Table 1. Using a lactate acid anaerobic threshold (AT) of 4 mmol/L, training intensity was classified as extensive (below AT), highly intensive (above AT), and extremely intensive (control tests). All camp participants exercised according to the same training plan and ate the same meals composed according to their energy requirements resulting from the training load. The analyzed training period began after 1 week of adaptation activities, which prepared the athletes to undertake regular training exercises, to reduce the undesirable impact of stress caused by changes in the participants’ environment on the measurement results. The test corresponded to one of the rowing competitions that athletes regularly compete in, which involves covering a distance of 2000 m in the shortest possible time. All participants took the tests used in the experiment many times before and had experienced the measurements involved, so they were completely habituated to them.

### 2.3. Experiment Protocol

The course of the experiment is presented in Figure 1a. The experiment was conducted in two sessions: at the beginning and the end of the training camp. Each of the sessions comprised four activities:

The first measurement of cortisol concentration at its initial level (I) at 7:00 a.m., before breakfast;an exertion stress test of 2000 m rowing at maximum intensity;the second measurement of cortisol concentration immediately after the exertion (E); andthe third measurement of cortisol concentration after restitution (R) on the second day at 7:00 a.m., before breakfast.

Before the first measurement, body composition was measured for group characterization.

### 2.4. Measurements

The concentration of free cortisol in the serum was determined. Peripheral venous blood was collected in a sample tube of the S-Monovette^®^ (Sarstedt, Nümbrecht, Germany) with a clot activator. The sample was then centrifuged at 4000 rpm for 10 min and frozen at 80 °C until determinations. Assays for the data collected over 2 months were conducted in one batch. The Euroimmun Analyzer I-2P (EUROIMMUN Medizinische Labordiagnostika AG, Lübeck, Germany) was used for the material examination. Cortisol concentrations were determined using a commercially available diagnostic test (ref. EIA-1887, DRG, Marburg, Germany). The coefficients of variation within and between tests were 7.2% and 5.4%, respectively.

Body composition measurements were taken using a Tanita MC980MA body composition analyzer (Tanita, Tokyo, Japan). Of the variables this device provided, body height and weight and body fat percentage were selected for statistical analysis.

### 2.5. Statistical Analysis

Statistical analyses were performed and developed with the R Language and Environment for Statistical Computing [28] software. Changes in cortisol concentration were estimated with the index of the relative change, according to the formula:(1)$$\mathrm{RC}\left(x_{1},x_{2}\right)\left[\%\right]=\ln\left(\frac{x_{2}}{x_{1}}\right)\cdot 100$$
where x_1_ is the value before the change, and x_2_ is the value after the change. The results were presented in terms of the description of the following variables:Cortisol concentration levels:
○I—initial (1st day, morning);○E—after the exertion test;○R—after restitution (2nd day, morning).Exertion-induced changes in cortisol levels:
○IE = RC(I, E)—immediate reaction to exertion;○IR = RC(I, R)—the expression of this reaction after recovery.
The training camp effects:
○∆IE = (IE_after_ − IE_before_)—change in immediate reaction to exertion;○∆IR = (IR_after_ − IR_before_)—change in the after-recovery expression of the reaction;○∆I = RC(I_before_, I_after_)—change in initial concentration levels;○∆E = RC(E_before_, E_after_)—change in concentration levels just after exertion;○∆R = RC(R_before_, R_after_)—change in concentration levels after recovery.

The statistical certainty of changes was estimated with the one-sample Cliff’s effect size (δ) in context of the predominance of their direction, the one-sample Cohen’s effect size index (d) in the context of mean values, and the *p*-value of a one-sample Welch’s *t*-test of the hypothesis of the zero-value of their mean. The strength of the relationship between the values of some selected descriptive variables was estimated with Kendall’s τ correlation index.

The data sets are described with their mean and 95% confidence intervals (CIs) presented in square brackets ([CI_lower_, CI_upper_]). The same way of describing the confidence interval was applied to the correlation index used. The results of the measurements and analyses are presented in Table 1 and are illustrated in Figure 1b,c. Additionally, Cliff’s δ effect sizes, representing the consistency in the direction of IE and IR, are presented in the text.

## 3. Results

The full set of estimates of changes in cortisol levels is included in Table 2. Six weeks of regular high-intensity training in elite rowers resulted in weakened cortisol concentration reactions to effort stress. Before the camp, cortisol levels consistently declined just after an exertion stressor (IEbefore ≈ −17%), and after a presumable subsequent rise, the next morning, it remained elevated (IRbefore ≈ 20%) relative to the first day’s morning value. After the camp, the post-exertion decline weakened markedly (IEafter ≈ −8.6%), and the elevation of the post-restitution level the next morning reduced to almost 0 (IRafter ≈ 1.4%). Thus, post-exertion and post-recovery cortisol levels significantly, albeit less consistently, approached baseline (Figure 1b), whereas the initial concentration (I) was a strong linear predictor of the post-exertion (E) and post-restitution (R) concentration (E = 0.90 · I, R^2^ = 0.95 and R = 1.065 · I, R^2^ = 0.95).

The sizes of above changes induced by training, independently of the initial levels, describe the indices of the sizes of immediate (ΔIE ≈ 7.1%) and post-restitution (ΔIR = −20.84%) manifestations of reactions to exertion (Figure 1c). As a result of eliminating the influence factor of the initial value common to both indices, their values measured before training did not correlate with their values after training (τ = 0.15 for IE and τ = −0.017 for IR). In addition, ΔIE and ΔIR correlated little and not very reliably (τ = 0.35), and ΔE and ΔR did not appear to correlate at all (τ = 0.083).

The training also resulted in a slight increase in the initial cortisol concentration (ΔI = 8.95%). This increase correlated positively (τ = 0.56) with the after camp (Iafter) but not with the before-camp (Ibefore) initial cortisol concentration levels (τ = −0.01). Likewise, because both E and R were positively correlated with I, the ΔE and ΔR were also positively correlated with cortisol values after camp but not before camp.

The ΔI-affected indices ΔE and ΔR, representing the overall adaptive changes in cortisol levels, showed the training effects more clearly than ΔIE and ΔIR (ΔE ≈ 19.1% and ΔR ≈ −13.8%).

Body fat percentage was strongly correlated with the training effect indices ΔIE (τ = 0.56, *p* = 0.003) and ΔE (τ = 0.53, *p* = 0.005). In addition, it was positively correlated with the after camp but not the before camp values of IE and E. The full set of correlation coefficients between body fat percentage and indices describing levels and changes of cortisol concentration is included in Table 3. The relationships between body fat percentage and initial and post-exertion levels of cortisol concentration and their training-induced changes are shown in Figure 2.

## 4. Discussion

This study investigated whether the decrease in cortisol levels after stress represents a physiological reaction or insufficiency of maintaining homeostasis. For this purpose, we analyzed how the cortisol concentration course changed in response to short but extremely high exertion in the process of adaptation to strenuous physical exercise training in healthy, highly trained elite athletes. Assuming that an immediate stressor-induced decline in cortisol triggers a counteracting of its deficiency, we hypothesized that adaptation to stress would result in a reduction in this deficiency. As we suspected, cortisol levels dropped just after extreme exertion stress and then increased during rest. We found that six weeks of high-intensity, efficient training weakened this decline, lowered the previously elevated cortisol level after rest, and slightly raised its initial morning level. We revealed that the post-stressor cortisol decline was reduced as a result of the adaptive process of improving the recovery of serum cortisol levels to maintain homeostasis. Additionally, we demonstrated that the HPA axis manages cortisol levels according to an anti-deficit principle.

Before the camp, participants consistently responded to the exertion test with lowered cortisol levels immediately after exertion (IE_before_) and elevated cortisol levels during rest (IR_before_) relative to the initial level (I_before_). In our view, this shows how works the not yet adapted to the level and form of the stressor HPA axis; the exertion-induced increase in cortisol consumption resulted in a temporary cortisol shortfall, which induced a late but exaggerated overproduction in response, counteracting this decline. After the camp, both the post-exertion cortisol (IE_after_) declines and the elevated cortisol concentrations after rest (IR_after_) became much smaller and less consistent, and the average cortisol concentration after rest (R_after_) returned to baseline (I_after_). The athletes maintained a level of performance not lower and a concentration of lactic acid not higher than that before the camp; this implies that the weakened reaction after a multi-day load of exhaustive exercise demonstrated the adaptation, allowing cortisol levels to remain within the normal range.

The weakening of the cortisol response to the exertion test resulting from camp training became apparent in the indices of cortisol level changes relative to morning cortisol values (∆IE and ∆IR) and a bit more prominent in the indices of overall training effects comprising changes in morning values (∆E and ∆R) that were non-relative to them. The training also resulted in a slight increase in initial morning values (∆I). A strong correlation of ∆I with the morning values after camp (I_after_) but not with the values before camp (I_before_) showed the possible independence of the adaptive effect of training on participants’ before-camp state. Regardless, the greater visibility of these changes in ∆E and ∆R showed that the elevation of morning values supports a counteracting stressor-induced cortisol deficiency. The higher morning cortisol concentration provided greater cortisol availability at the onset of stress at the expense of only a slight slowdown in restoring its previous day’s level during rest. Taken together, this suggests the existence of two components of adaptation counteracting cortisol deficiency: first, adjusting the speed of cortisol concentration changes to altered functional conditions and, second, modifying baseline cortisol levels. Hence, adaptation to stress follows the principle of avoiding cortisol deficiency in at least two ways: speeding up the response to the decline in cortisol for rapid restoration of its previous levels and increasing the baseline cortisol concentration for greater cortisol availability at the onset of stress. Based on a simple mathematical transformation, one can also show that the sum of the values of indices of these two components (∆I and ∆IE) gives the value ∆E (∆E = RC(E_before_, E_after_) = ∆I + ∆IE). Likewise, ∆R = RC(E_before_, E_after_) = ∆I + ∆IR.

We further noted that the complementarity of the two components counteracting the cortisol deficiency after the onset of stress exhibited wide individual variability, despite the athletes’ similar fitness, performance, and body build, as well as their standard diet and training pattern during the camp. At the beginning of the camp, all participants reacted consistently to the exertion test, showing a common pattern of reaction to the stressor. After the camp, this reaction varied much more between individuals. This suggests that the functioning of the HPA axis is subject to a wide range of factors in personal individualization, which manifests in variation in the timing and level at which cortisol concentrations change around exercise. This is supported by the relationships of ∆I, ∆IE, and, consequently, ∆E values with body fat percentage presented in this study (Figure 2, Table 3). In the overall training effect, counteracting post-exertion cortisol deficiency improved more in participants with a higher body fat percentage. This may result from the more rapid acquisition of cortisol from adipose tissue than from other sources. Thus, the individualization of the response to a stressor, without excluding other factors, stems to some extent from the rate at which individuals can obtain cortisol to counteract the cortisol deficit at the onset of stress. This complex system of interdependencies, affecting both the ability to adapt to stress and the current state of adaptation, explains why, despite the observable phenomenon of the general rule of counteracting cortisol decline, the effect size of training cannot be predicted from the size of the cortisol deficit.

Our study reports for the first time the anti-deficit principle of the HPA axis response to a stressor. The finding that accelerating the response to counteract early stressor-induced cortisol deficiency reduces the levels of cortisol excess resulting from a delay in counteracting this deficiency accords with previous views on the role of the HPA axis in protection against hypercortisolemia. However, early activation of counteracting cortisol deficiency at the onset of stress requires the release of cortisol from stores much earlier than the pituitary gland could trigger alone; it must thus take place via a non-neuronal pathway. Reports on maintaining cortisol responsiveness in rats devoid of the hypothalamus [18] and the short loop of the HPA axis [19] have considered this possibility. The positive correlation between the magnitude of the stress response and body fat percentage observed in previous studies further supports this view while suggesting that the regeneration of cortisol stores is more straightforward with the use of body fat stores [29]. Our results also confirm this observation by reporting greater cortisol concentration changes in response to a stressor in individuals with a higher body fat percentage, demonstrating the extent to which regeneration of cortisol stores takes place in the adipose tissue; this supports knowledge of the nutritional determinants of the efficacy of training-induced adaptation [30]. The adaptive effect of increasing the baseline cortisol concentration was already reported in a study on eight-week intensive training of athletes [31]. Presented in our work, complementary ways of compensating for cortisol deficiency ultimately confirm that adaptation to stress aims to increase cortisol availability in a case of a stressor.

The immediate post-exertion decline in the cortisol level observed in our study has also been reported in several studies showing a similar reaction following moderate- to high-intensity exercise sessions [4,6]. The authors of these studies did not expect this type of reaction and thus referred to them as an “unexpected lack of desired increases” [5]. However, our study reveals the rationale behind treating them as normal physiological phenomena. In our opinion, this decline manifests a significant growth of this hormone’s consumption after exposure to a stressor, particularly evident in the absence of adaptation to stress. This has been shown, for example, for exam-related tension [5], physical exhaustion in athletes [4], and significant training loads in postmenopausal or elderly people. In the case of older adults, research has explained the unexpected decline in cortisol level by an insufficient intensity of exertion applied despite the use of submaximal efforts [7,8]. We believe that in this case, the reason was the opposite: the load may not have been too large when treated as unconditional, but it was large in respect to the current individual efficiency of coping with this load. In our study, the unambiguous capture of the stressor-induced cortisol decline resulted primarily from the application of an exertion stressor of an intensity pushing the limits of human capabilities, which was extremely high even for the most well-trained elite athletes. The decrease thus appears to result not from an insufficient but rather an excessive strength of the stressor relative to the current state of adaptation, conditioned in turn by several factors, including age. Vulnerability to these factors may have caused the inconsistencies in the assessments from the analyses presented in earlier research papers. Differences between participant groups studied in the earlier works may be present not only because of different types and intensities of exercise, but also because of different adaptive predispositions. The inadequacy of the strength of the stressor for the other determinants of adaptation efficiency may also explain, to some extent, why the observability of cross-stressor adaptation among non-athletes faces difficulties. Activation of the adaptation processes requires inducing a sufficiently large decrease in cortisol concentration by the application of a sufficiently large stressor. It is likely, therefore, that in individuals in whom safety considerations do not allow the application of acute exertion, cortisol levels at the onset of stress do not decrease sufficiently to induce an adaptation phenomenon pronounced enough to become visible against the background of the impact of other individual determinants of the effectiveness of this adaptation.

The weakening of the cortisol response to an exertional stressor in healthy athletes following prolonged efficient training presented in this study corresponds with the results of other works, which refer to the attenuation of the amplitude of cortisol concentration changes as a “slower rate of increase and lower peak, and speedier recovery back” [32], “reduced fluctuations” [33], and “blunting of changes” [16]. By demonstrating that adaptation to exertion-caused stress results in a lowered magnitude of cortisol concentration changes, we support the theses presented in some works [34,35] that this attenuation results from increased efficacy of counteracting cortisol deficiency due to the increased sensitivity of the cortisol regulatory system. Admittedly, the authors of some studies noted this phenomenon mostly in individuals affected by OTS and, therefore, are inclined to consider OTS the cause of this attenuation, attributing such results to the impaired sensitivity of the HPA axis [4,33]. However, the results of our experiment demonstrate that in healthy individuals adapted to stress who show no signs of OTS, this phenomenon also occurs. Thus, this weakening cannot result from OTS but, at most, accompanies it. Most likely, the attenuation of cortisol changes in response to a stressor, often accompanying OTS symptoms, reflects a state of high adaptation to the frequent stressors experienced, both before the formation of this syndrome and later in its course.

A limitation of our study stems from the experimental advantage of the homogeneity of the study group: all participants showed a positive attitude toward effort and psychological tolerance for extreme exhaustion, trusting in its benefits and experiencing a sense of self-efficacy. Thus, at the expense of the possibility of thorough observation of the phenomena analyzed, we could not estimate the possible impact of differences in attitude on the way experimental participants reacted to the stressor, which may strongly mediate the adaptation to stress. Acceptance of exhaustion may play a crucial role in undertaking further training loads and achieving adaptation. The issue of achieving this acceptance in unmotivated or less mentally resilient individuals constitutes a separate problem for further scientific consideration. On the other hand, on the basis of current research, one can speculate about achieving the induction of a cortisol deficit with a stressor with less exhausting consequences. Thus, in non-adapted individuals, one could more easily apply cross-stressor adaptation in therapeutic practice with an exertion that is objectively smaller but large relative to their health capacity. Severe fatigue means reaching one’s limits; hence, the efficiency of adaptation may have important implications in the awareness of the beneficial effects of experiencing this fatigue. Thus, the task of developing a positive attitude toward exposure to stressors during adaptational training seems to constitute the relevant component of training procedures [36], which is especially socially important for people with a weakened mental and physical condition because it may prevent the development of stress-related pathologies associated with aging and improve quality of life [37]. This suggests the need to create a framework for overcoming individual current limits in the face of trainees’ beliefs in their insufficiency through controlled exercise training.

## 5. Conclusions

The decrease in cortisol at the onset of stress results from the temporary inability of the cortisol regulatory system to keep up with the increased demand for cortisol caused by an increase in stress level. The resulting cortisol deficit provides the stimulus for the stress adaptation that leads to the attenuation of reactive changes in cortisol concentration. Therefore, one should regard the stress-induced cortisol decrease as. in principle. a normal physiological phenomenon, not necessarily accompanying mental or physical impairment.

Adaptation to stress follows the principle of counteracting cortisol deficiency in the face of the stressor. The body improves the effectiveness of avoiding this deficiency by increasing baseline cortisol concentrations for greater cortisol availability at the onset of stress and by speeding up the response to the decline in cortisol for rapid restoration of its previous levels with the support of adipose tissue. This adaptive improvement in the precision of the response of the cortisol regulatory system leads to an attenuation of stress-induced changes in cortisol concentrations. Because of the adaptive reduction in the amplitude of the stressor-induced deficiency, the subsequent cortisol surplus, which persists long after the stressor has subsided, decreases, maintaining homeostasis.

Achieving the effectiveness of adaptive training requires inducing a substantial temporary cortisol deficit by stimulating a stressor strong enough to cause a temporary demand for cortisol greater than the body’s ability to satisfy it consistently at the onset of stress. The attenuation of changes in cortisol concentration in response to a stressor observed in the diagnosis of certain disorders does not necessarily result directly from these disorders because it may merely represent a possible manifestation of adaptation to regularly experienced stressors accompanying these disorders.

## Figures and Tables

**Figure 1 ijerph-19-05633-f001:**
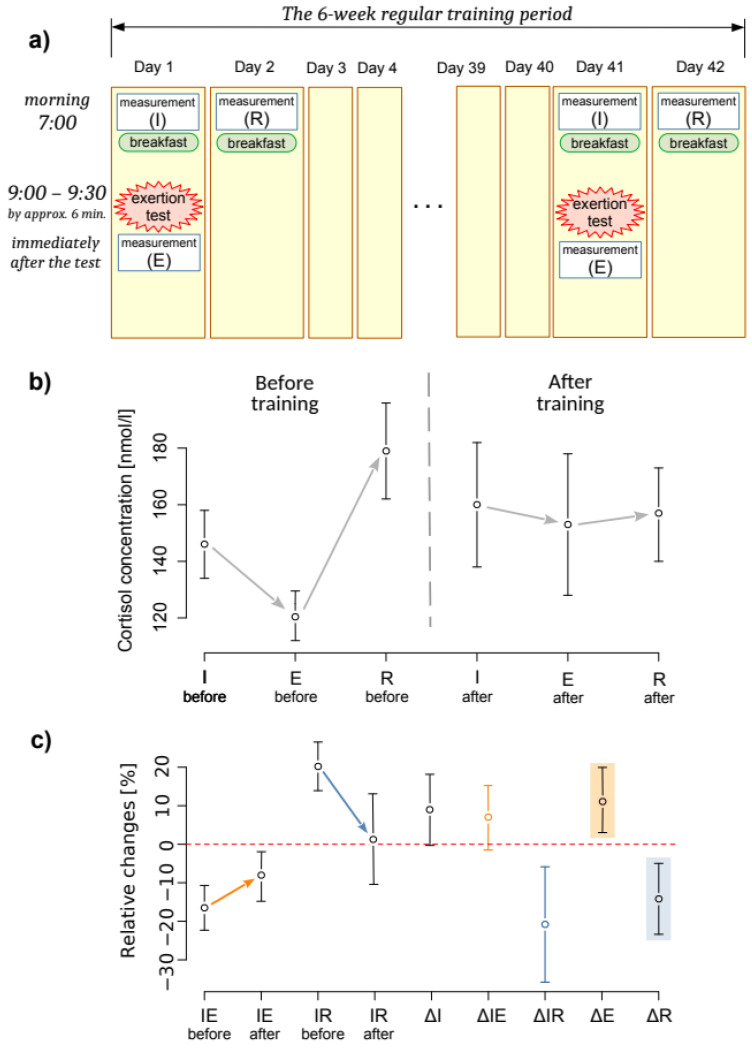
The experimental protocol (**a**), the training-derived changes in cortisol concentrations shown by the mean values of cortisol concentrations (I, E, R) (**b**) and the relative changes representing the reaction to the test (IE, IR) and training effect (ΔI, ΔIE, ΔIR, ΔE, ΔR) (**c**), along with their 95% confidence intervals. The test-induced reactions approached zero (weakened) as the IE value increased (yellow arrow and point-and-whiskers) and the IR value decreased (blue arrow and point-and-whiskers).

**Figure 2 ijerph-19-05633-f002:**
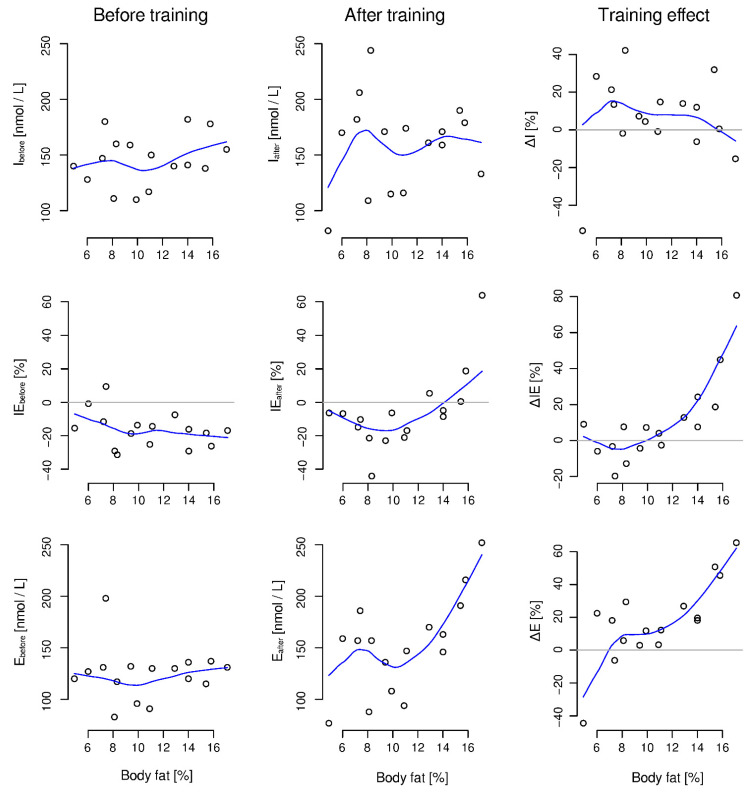
Relationships between body fat percentage and cortisol concentrations before and just after the exertion test, along with their changes due to the test and the training. Participants with a higher body fat percentage showed a more pronounced training effect in terms of a flattening of the immediate post-exertional drop in cortisol concentration (ΔIE) and a much weaker, faintly evident increase in morning values (ΔI). Thus, a higher percentage of body fat was associated with a more pronounced overall training effect in terms of the increase in post-exertional cortisol concentration (ΔE).

**Table 1 ijerph-19-05633-t001:** Distribution of increasing training load over the course of training and the value of lactic acid (LA) in the study group before and after the exertion test.

	1st Week			Last Week		
	Training times [min/Day]		Days per Week	Training times [min/Day]		Days per Week
Type of Exertion	Mean	SD		Mean	SD	
Training for force development	80.00	14.14	2	75.00	7.07	2
Extensive endurance rowing	77.00	17.31	7	78.83	17.09	6
High-intensity endurance rowing	25.25	10.50	4	87.86	16.66	7
Very high-intensity endurance rowing	0	0	0	31.67	11.59	3
Unspecific training (running, among others)	26.43	24.10	7	22.86	25.63	7
	LA concentration [mmol/L]			LA concentration [mmol/L]		
Lactate acid measurement	Mean	SD		Mean	SD	
Before exertion test at the end of a week	1.43	0.20		1.64	0.57	
After exertion test at the end of a week	16.42	3.43		15.05	3.02	

**Table 2 ijerph-19-05633-t002:** Results of the experiment.

		Before Training		After Training	
		Mean	SD	Mean	SD
Cortisol concentration [nmol/L]	I	146.0 [133.9, 158.1]	22.78	160.1 [138.4, 181.8]	40.69
	E	121.2 [112.0, 130.4]	17.31	151.9 [128.2, 175.6]	44.51
	R	176.9 [163.6, 190.2]	24.97	156.7 [140.2, 173.3]	31.14
Reaction to exertion (%)	IE	−16.98 [−22.16, −11.79]	9.73	−8.59 [−14.92, −2.26]	11.87
	IR	19.87 [13.95, 25.80]	11.12	1.42 [−10.36, 13.20]	22.11
Training effects		Mean (%)	Cohen’s d	Cliff’s δ	*p*
	ΔI	8.95 [−0.18, 18.09]	0.52 [−0.01, 1.05]	0.37 [−0.17, 0.74]	0.054
	ΔIE	7.14 [−1.27, 15.56]	0.45 [−0.08, 0.98]	0.25 [−0.28, 0.66]	0.090
	ΔIR	−20.84 [−35.83, −5.85]	−0.74 [−1.27, −0.21]	−0.5 [−0.82, 0.05]	0.010
	ΔE	19.07 [8.33, 29.82]	0.95 [0.41, 1.48]	0.75 [0.21, 0.94]	0.002
	ΔR	−13.79 [−23.02, −4.46]	−0.80 [−1.33, −0.26]	−0.5 [−0.82, 0.05]	0.006

**Table 3 ijerph-19-05633-t003:** Correlation between body fat percentage and cortisol concentrations (I, E, R) and reactions to exertion (IE, IR) for the states before training and after training and the changes in these values between both states, which resulted from training.

	Before Training		After Training		Training Effect (Δ)	
	τ	*p*	τ	*p*	τ	*p*
I	0.15 [−0.25, 0.51]	0.417	0.10 [−0.30, 0.47]	0.588	−0.16 [−0.51, 0.24]	0.392
IE	−0.23 [−0.56, 0.17]	0.224	0.41 [0.03, 0.69]	0.027	0.56 [0.22, 0.78]	0.003
IR	0.06 [−0.33, 0.43]	0.752	0.02 [−0.36, 0.41]	0.892	0.07 [−0.32, 0.45]	0.685
E	0.14 [−0.26, 0.50]	0.470	0.42 [0.04, 0.70]	0.024	0.53 [0.18, 0.76]	0.005
R	0.15 [−0.25, 0.51]	0.417	0.13 [−0.27, 0.49]	0.498	0.09 [−0.3, 0.46]	0.620

## Data Availability

The datasets generated during and/or analyzed during the current study are available from the corresponding author on reasonable request.

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
