# Peer review of "Stressor-Induced Temporal Cortisol Deficiency as a Primary Trigger for Adaptation to Stress"

_ijerph, 2022, doi:10.3390/ijerph19095633_

Round 1
Reviewer 1 Report
The study of Ewa Latour and colleagues was conducted on 16 professional athletes and that was it’s main drawback. The small sample size and professional characteristics of the participants do not allow one to make the fully justified conclusions. The authors did not compare the results with the sedentary controls matched by age. Otherwise, if the studied group include only athletes, they should be tested during pre-competition (training), competition (trainig) or out-of competition period, as their training rhythm affects the normal circadian rhythm of cortisol secretion by leading to a phase shift HPA axis activity (Van Reeth O, et al. Am J Physiol. 1994 Jun;266(6 Pt 1):E964-74). Moreover, according to AC Hackney and A Viru (1999), the substantial and persistent nocturnal suppression in the hormone levels in response to transient elevations in cortisol during daytime exercise can be observed. This fact can also influence the main thesis of the study.
Based on these I have some concerns
The major concerns.
- The population studied is too small and homogenous and thus the contribution to the variability in HPA axis reactivity may be the cause of the identified patterns, since it is known that both animals and people are non-heterogeneous in response to stress and differ in both hormonal response and neurotransmitter metabolism (Tseilikman et al. Psychoneuroendocrinology. 2020 Jul;117:104691; Ponzi D, et al. Adaptive Human Behavior and Physiology. 2022) . No sample size calculations. No sedentary controls. The selection effect of the study makes its difficult to identify general patterns, as the athletes have significantly higher levels of cortisol in comparison to general population (Popovic B, et al. J Med Biochem. 2019 Mar 1;38(1):6-12.).
- There is no baseline data in pretraining period (I strongly doubt that athletes train with such intensity all year round).
- The title of the manuscript should be changed to clarify the type of the study and the studied population. The presented title is too generous.
- I would like to suggest shortening the introduction and discussion for better reading and to make it more focused on the exact goals of the study.
Minor concerns. Probably you should consider to use the concept of allostasis (process through which various biological processes attempt to restore homeostasis when an organism is threatened by various types of stress in the internal or external environment) and allostatic load, but not the adaptation, as some of the passages in the text seems to be generalized and not very precise in terms of pathophysiology. Moreover, there is a concept of maladaptation, which could more accurately reflect some aspects.
Author Response
"Please see the attachment."

Reviewer 2 Report
An interesting and well design paper on cortisol modifications in stress adaptation in some specific athletes. The paper is ok and of interest for the readers in the controversial area of cortisol variations. Introduction does set the scene quite well. Methology is ok, but I do have a problem with the fact that in abstract the subjects are named “successful” or something at their sports. Even elite athletes is a relative term. This is of course highly subjective and either way no relevant data on this is offered. Please make this clear on the methodology. Focusing on the 7 years average of practicing this sport might be a better option. Discussion is quite well balanced and brigs classical aspects into light. Good work in general.
Author Response
We are very satisfied with your high evaluation of our work.
In fulfilment of your recommendations, we have corrected the relevant parts of the description in the Abstract and Methodology section.
In Abstract section, we have changed the text:
"We examined 16 male elite endurance athletes during their training camp before the World Rowing Championships, who were successful in these competitions."
to:
"We examined 16 male athletes, members of the Polish National Rowing Team, competing in the World Rowing Championships in top form, of average training experience of 7 years. The measurements we performed before- and after the training camp preparatory to the Championships. ".
In Material and Methods section, we have changed the text:
"All the participants were members of the Polish Rowing Team, and the study was conducted during their preparation training camp before the World Rowing Championships. Their mean training experience was 7 (SD = 1.71) years."
to:
"The participants were members of the Polish Rowing National Team. They were selected for the Team based on their performance in national rowing competitions and the results of performance tests corresponding to the rowing competitions. After a preparatory training camp, the national team members competed at the World Rowing Championships in peak form. The study group comprises the members who agreed to take measurements related to the research. Their mean training experience was 7 (SD = 1.71) years".
Reviewer 3 Report
The manuscript addresses an interesting topic, the results of the research can be partially considered innovative.
Research questions / hypotheses are indicated in the introductory passage, but perhaps they deserve a clearer definition.
The methodological procedures are described, but a more precise description would also be needed, among other things, it is not even stated in which year the study was conducted, only the months are given.
A relatively fundamental shortcoming is a very small research sample - 16 participants. I miss the indication of the method of selection of participants and also the method of selection, but above all the choice of a given number.
The results are documented by a number of tables and graphs, including a very extensive text in which the reader is "easily lost". The passage of results would need to be systematized, described more carefully.
The discussion is extensive, perhaps acceptable.
I do not consider the conclusion to be well drawn, I recommend reworking it, clearer, more emphatic formulations would benefit the whole work.
I recommend that you carefully review and edit your English if the article is accepted for publication.
Author Response
We are pleased with your generally good opinion of our work.
We also found the sample size not very impressive, but fortunately the dispersion of the data obtained was small enough that, with this sample size, the confidence intervals proved to remain sufficiently narrow with respect to the mean values to statistically validate our results.
Guided by your comments and helpful suggestions, we have made the following changes:
We have refined the paragraph introducing the hypothesis.
We have improved the description of study group with the information about a rationale and the method of selection participants to the group, that also justifies the resulted sample size. Additionally, we have completed the information about the year of the study.
We have improved the readability of the textual description of the results by replacing the detailed numerical descriptions of the values of the indicators, which we present in the tables anyway, with their approximate values. In addition, we have also converted the numerical description of cortisol levels presented in table 2 for compatibility with SI units' system so that they are expressed in [nm/L].
We have also refined the first paragraph of the 'Conclusions' section to make it more explicit and clearer.
Reviewer 4 Report
The article "Stressor-induced temporal cortisol deficiency as a primary trigger for adaptation to stress" submitted for review contains an interesting analysis of cortisol response to top-intensity physical exertion. The manuscript contains a well-developed theoretical part. The research part was properly designed and comprehensively described. The obtained results were thoroughly analysed and discussed. Interesting conclusions deserve attention.
Formal comments:
- Please prepare the affiliation according to the guidelines for authors („complete address information including city, zip code, state/province, and country”).
- Please prepare references according to the guidelines for authors.
Author Response
We are very satisfied with your high evaluation of our work.
In fulfilment of your recommendations, we have added the full address of our affiliation and corrected the format of the references.
Round 2
Reviewer 1 Report
Dear authors, thank you for the conducted work. It was a great pleasure for me to read your answers. I even got the feeling that we are at a symposium on stress disorders. I wish you success!
Reviewer 3 Report
The authors made adjustments to the original manuscript, which contributed to improving the quality of the article, and I recommend the manuscript for publication.